# Maternal Selenium, Copper and Zinc Concentrations in Early Pregnancy, and the Association with Fertility

**DOI:** 10.3390/nu11071609

**Published:** 2019-07-16

**Authors:** Jessica A. Grieger, Luke E. Grzeskowiak, Rebecca L. Wilson, Tina Bianco-Miotto, Shalem Y. Leemaqz, Tanja Jankovic-Karasoulos, Anthony V. Perkins, Robert J. Norman, Gus A. Dekker, Claire T. Roberts

**Affiliations:** 1Robinson Research Institute, University of Adelaide, 5005 North Adelaide, South Australia, Australia; 2Adelaide Medical School, University of Adelaide, 5005 Adelaide, South Australia, Australia; 3Current affiliation: Center for Fetal and Placental Research, Cincinnati Children’s Hospital and Medical Center, Cincinnati, OH 45229, USA; 4Waite Research Institute, School of Agriculture, Food and Wine, University of Adelaide, 5064 Adelaide, South Australia, Australia; 5School of Medical Science, Griffith University, Gold Coast Campus, 4215 Southport, Queensland, Australia; 6Fertility SA, 5000 Adelaide, South Australia, Australia; 7Women and Children’s Division, Lyell McEwin Hospital, University of Adelaide, 5112 Adelaide, South Australia, Australia

**Keywords:** time to pregnancy, trace elements, subfertility, fertility, selenium, zinc, copper, pregnancy

## Abstract

Trace elements such as zinc, copper, and selenium are essential for reproductive health, but there is limited work examining how circulating trace elements may associate with fertility in humans. The aim of this study was to determine the association between maternal plasma concentrations of zinc, copper, and selenium, and time to pregnancy and subfertility. Australian women (*n* = 1060) who participated in the multi-centre prospective Screening for Pregnancy Endpoints study were included. Maternal plasma concentrations of copper, zinc and selenium were assessed at 15 ± 1 weeks’ gestation. Estimates of retrospectively reported time to pregnancy were documented as number of months to conceive; subfertility was defined as taking more than 12 months to conceive. A range of maternal and paternal adjustments were included. Women who had lower zinc (time ratio, 1.20 (0.99–1.44)) or who had lower selenium concentrations (1.19 (1.01–1.40)) had a longer time to pregnancy, equivalent to a median difference in time to pregnancy of around 0.6 months. Women with low selenium concentrations were also at a 1.46 (1.06–2.03) greater relative risk for subfertility compared to women with higher selenium concentrations. There were no associations between copper and time to pregnancy or subfertility. Lower selenium and zinc trace element concentrations, which likely reflect lower dietary intakes, associate with a longer time to pregnancy. Further research supporting our work is required, which may inform recommendations to increase maternal trace element intake in women planning a pregnancy.

## 1. Introduction

Impaired fertility, the failure to achieve pregnancy after 12 months or more of regular unprotected sexual intercourse, afflicts millions of couples worldwide, placing a significant emotional and economic burden on themselves, their families, and society [1]. There are consistent behavioural, clinical and biological factors associated with infertility including poor dietary intake, cigarette smoking, advanced maternal age, obesity, and polycystic ovary syndrome. Additionally, there is emerging, albeit inconsistent, evidence to suggest that high dose and occupational level exposures to toxic metals such as mercury, cadmium, and lead, associate with reduced fecundity and fertility in couples conceiving naturally [2,3] or requiring assisted reproduction [4,5,6,7]. Importantly, these elements are not essential to human health.

Comparatively, trace elements such as zinc, copper, and selenium are essential for health and are necessary for reproductive health [8,9]. There is considerable evidence in pregnancy highlighting their effects on oxidant/antioxidant balance, but also their other roles such as cell proliferation (zinc), protein synthesis (selenium, zinc), and haematopoiesis (copper) [9]. There are established roles for selenium, zinc, and copper in relation to male fertility [10,11], but there is far less known about trace element concentrations and female conception. In a sample of 80 pregnancy planners, no association was found between plasma zinc and the likelihood for a positive pregnancy test [12]. In contrast, in 45 women undergoing in vitro fertilisation (IVF), higher urinary copper (≥35.7 μg/L) was associated with higher total number of oocytes retrieved and better embryo quality, and higher urinary copper and zinc (≥1808 μg/L) was associated with total number of embryos generated [13]. Further, women who had unexplained infertility had lower selenium in their follicular fluid, the microenvironment which nourishes and surrounds the oocyte, compared to women with tubal infertility and male factor infertility [14]. In addition, recent studies have demonstrated both positive and negative associations between selenium, zinc and copper in the follicular fluid of women with and without endometriosis, which is a common cause of infertility [15,16]. Overall, these studies were limited by small sample sizes or were in women undergoing IVF or in women who had fertility disorders and so might not be representative of the general population, making the relationship between certain trace elements and fertility unclear. A major challenge in evaluating the effects of certain exposures on fertility lies in recruiting a sufficiently large sample of women prior to pregnancy and following them prospectively. Retrospective time to pregnancy studies which recruit women during pregnancy and ask them to recall exposures prior to conception and the time taken to conceive, have emerged as a valid and informative epidemiological approach to better understanding factors influencing female fertility [17,18]. One limitation of this approach lies in the required assumption that data collected following conception, whether clinical or biological, is reflective of pre-conception exposure status. However, we have previously utilised this to investigate the effects of various factors such as pre-conception dietary intake, asthma medication use, or maternal metabolic health status on time to pregnancy [19,20,21]. Further, having previously assembled a large cohort of women to investigate the effects of early pregnancy trace element status on pregnancy outcomes [22], and knowing that previous research has demonstrated a strong correlation between early pregnancy and pre-conception concentrations of trace elements such as selenium [23], we were in the unique position to investigate the relationship between these factors and reported time to pregnancy.

The aim of this study was to determine the association between maternal plasma concentrations of the trace elements, zinc, copper, and selenium, and time to pregnancy and subfertility. We hypothesise that lower concentrations of trace elements will associate with longer time to pregnancy and a higher rate of subfertility.

## 2. Materials and Methods 

### 2.1. Study Population

The Screening for Pregnancy Endpoints study (SCOPE) is a multi-centre prospective cohort study which recruited nulliparous women with singleton pregnancies from Adelaide (Australia), Auckland (New Zealand), Cork (Ireland), Leeds, London and Manchester (UK) (*n* = 5628). Data for this study was from the Adelaide cohort (*n* = 1164). Ethics approval was gained from the University of Adelaide ethics committee and all women provided written consent (approval no: REC 1712/5/2008).

Nulliparous women carrying a singleton pregnancy were recruited at 14–16 weeks’ gestation from the Lyell McEwin Hospital, Adelaide, Australia between November 2004 and September 2008. Research midwives collected information on demographics, smoking, family, medical and gynecological history, diet and supplement use, height, weight, systolic and diastolic blood pressure, and waist circumference. A non-fasting plasma sample was obtained for biochemical measurement of copper, zinc, selenium and C-reactive protein (CRP). Exclusion criteria for the SCOPE study were women considered to be at high risk of preeclampsia, small for gestational age or preterm birth, those who previously had a cervical knife cone biopsy, had three or more pregnancy terminations or miscarriages, if their pregnancy was complicated by a known major fetal anomaly or abnormal karyotype or if they received interventions that may modify pregnancy outcome (e.g., aspirin, cervical suture), if they were taking high dose supplements, or had diabetes (type 1 or type 2) or hypertension and related disorders/treatment.

For this analysis, we excluded women where data on time to pregnancy was missing (*n* = 5) or where blood samples were unavailable for trace element analysis (*n* = 99), leaving a final cohort of 1060.

### 2.2. Trace Element Samples

Non-fasting plasma was obtained from heparinised, venous blood samples taken at 15 ± 1 weeks’ gestation. In order to measure trace element concentrations, plasma samples were digested in concentrated nitric acid (~70% HNO_3_) under pressure and at temperature as per our previous publication [22]. Samples were run alongside two internal standards: iridium and rhodium (Choice Analytical) at a concentration of 200 ppb and an 8-point calibration, including blank, was carried out between 0.01 μg/L and 100 μg/L. Copper, selenium and zinc concentrations were determined using inductively-coupled plasma mass spectrometry (ICP-MS) (Agilent 7700 ICP-MS; carried out by accredited CSIRO Analytical Services, South Australia). Laboratory surfaces were thoroughly cleaned to avoid contamination and all procedures were performed under conditions that ensured no trace mineral contamination. Samples were run in triplicate through the mass spectrometer. Concentration categories for each trace element were made in accordance with published pregnancy and laboratory studies values for second trimester of pregnancy, with reference points being the middle concentration ranges: copper (<25.97 µmol/L, ≥25.97 to ≤34.78 µmol/L, >34.78 µmol/L); zinc (<7.80 µmol/L, ≥7.80 to ≤12.24 µmol/L, >12.24 µmol/L); selenium (<0.95 µmol/L, ≥0.95 to 1.84 µmol/L, >1.84 µmol/L) [24]. However for selenium, no woman had a value over 1.84 µmol/L, thus the categories were amended to: <0.95 µmol/L vs ≥0.95 µmol/L [as the reference value]). To convert from SI units (µmol/L) to conventional units, the following equations were used: zinc (µmol/L × 6.53 µg/dL); selenium (µmol/L × 78.74 µg/L); copper (µmol/L × 6.37 µg/dL). In supplementary analysis, we categorised trace element concentrations according to tertiles, based on study population values.

### 2.3. Assessment of Outcome

Estimates of reported time to pregnancy (TTP) were derived from the following question “duration of sex without contraception before conception with father of baby”. Values in months were recorded in a continuous fashion and used to define TTP. In this study, subfertility was defined as having a TTP of more than 12 months or use of assisted reproductive technologies. Women who conceived in their first month had a reported TTP of 1 month, with prolonged TTP censored at 12 months of attempting to conceive.

### 2.4. Assessment of Covariates

Maternal ethnicity was self-reported and binary coded as Caucasian or other (90% were Caucasian). The socioeconomic index was developed in New Zealand and is a measure of the individual’s socioeconomic position derived from a specific occupation [25]. It provides a value of 10–90 with a higher score indicating higher socioeconomic position [26]. Retrospectively reported data at this same visit was also obtained: cigarette use and any intake of alcohol per week in the 3 months pre-pregnancy were each binary coded as yes or no. Dietary intakes relating to the one month preceding conception was obtained using single item questions of specific foods. The use of single item questions has been shown to be useful to assess gross level estimates and rank individuals on intakes, rather than precise levels of intake [27]. Number of servings of foods was reported to the midwife for fruit (fresh fruit and fruit juice), green leafy vegetables (vegetables high in folate such as spinach, cabbage, lettuce, broccoli), fish (with prompts of fish such as salmon, trout, sardines, shellfish and shrimp) and key discretionary foods reported to be consumed from take-away or fast food outlets (i.e., frequency of intake of burgers, fried chicken, pizza and hot chips were totalled as ‘fast food’). Frequency of fruit was categorised as ≥3 times/day vs. <3 times/day; green leafy vegetables as ≥1 times/day vs. <1 time/day; fish as ≥1 time/week vs. <1 time/week; fast food as never vs. more than never [20]. Frequency of sexual intercourse was described and reported as the frequency of sexual intercourse, per month, in the three months prior to conception, with the presumed biological father of the baby. Intake of multivitamin supplement in the first trimester (yes/no) was determined as either a multivitamin tablet containing ≥2 vitamins or ≥2 single vitamin tablets or a single vitamin with folate at the same time point (e.g., vitamin C tablet plus folate supplement). Adjustments were also made for whether the multivitamin included the particular trace element of interest (i.e., in analyses assessing zinc concentrations, we adjusted for whether zinc (yes/no) was contained in the multivitamin). Paternal data included age, height, and weight, and was self-reported from the biological father.

### 2.5. Statistical Analyses

Frequencies and descriptive statistics of all women were expressed as n (%) or as means (standard deviation, SD). Median (inter-quartile range, IQR) was reported when continuous variables were not normally distributed. The impact of each trace element (in categories) on TTP was investigated using accelerated failure time models, as previously published [19,20] with log normal distribution to estimate time ratios (TR) and 95% CIs. Adjusted marginal estimates for median TTP according to trace element concentrations were calculated using the Stata ‘margins’ command, together with corresponding 95% CIs. Alternative distributions were investigated but the log normal distribution was selected based on providing the lowest −2 log likelihood and Akaike information criterion value. These TRs can be interpreted as the ratios of the median values of the duration (in months) to achieve pregnancy between the compared groups. A TR above 1 implies that a given exposure is associated with longer TTP, whereas a TR below 1 indicates a shorter TTP.

We used causal diagrams (directed acyclic graphs) to guide selection of potential confounders for which to control, based on a priori selection of variables associated with trace element concentrations and fertility. Three separate adjusted models were used. Model 1: maternal age, maternal body mass index (BMI), ethnicity, socioeconomic status, plasma CRP, pre-pregnancy alcohol consumption, pre-pregnancy smoking status, frequency of sexual intercourse prior to conception, multivitamin use in first trimester, and trace element of interest in multivitamin; Model 2: Model 1 plus pre-pregnancy intake of fast food, green leafy vegetables, fruit, and fish; Model 3: Model 2 plus paternal age and paternal BMI. This was a complete case analysis.

Statistical significance was defined as a two-sided *p*-value of <0.05. All statistical analyses were undertaken using Stata IC 14 (Stata, College Station, TX, USA).

## 3. Results

### 3.1. Participant Characteristics

Participant characteristics are reported in Table 1. In total, 1060 women were included of whom 894 (84.3%) took <12 months to conceive, and 166 (15.7%) took 12 months or more. Women who took longer to conceive tended to have a lower frequency of sexual intercourse, a lower percentage of consumed alcohol prior to conceiving, a lower percentage of consumed fruit, ≥3 times per day, a higher percentage of consumed fish, ≥1 time per week, and a higher percentage of the women consumed a multivitamin in the first trimester.

### 3.2. Relationship between Trace Elements and Time to Pregnancy

The relationship between each trace element concentration at 15 ± 1 weeks’ gestation and TTP is presented in Table 2. Compared to the reference value, women who had lower selenium concentrations (<0.95 µmol/L vs. ≥0.95 µmol/L) had a longer TTP (adjusted TR, 1.19 (1.01–1.40)), corresponding to a median 3.2 vs. 2.6 months to conceive (Figure 1). Lower (<7.80 µmol/L) but not higher (>12.24 µmol/L) zinc concentrations were also associated with a longer time to pregnancy (adjusted TR, 1.20 (0.99–1.44)). Marginal estimates for median TTP in women with lower vs. normal zinc concentrations was 3.3 months vs. 2.7 months, respectively (Figure 1). There were no associations between copper concentrations and TTP. Excluding women requiring assisted reproductive technology (ART), Appendix A shows lower zinc concentrations (<7.80 µmol/L vs. ≥7.80 to ≤12.24 µmol/L) were associated with longer TTP (adjusted TR, 1.21 (1.01–1.44)). In all women, Appendix A shows similar time ratios when using study population zinc tertiles rather than laboratory reference ranges. For selenium, higher levels (>0.97 µmol/L vs. ≥0.86 to ≤0.97 µmol/L) were associated with a shorter TTP (TR 0.80 (0.65–0.99)).

### 3.3. Relationship between Trace Element Concentrations and Subfertility

Categories of trace element concentrations and risk for subfertility are presented in Table 3. Women with low selenium concentrations (<0.95 µmol/L vs. ≥0.95 µmol/L) were at a 1.46 (1.06–2.03) greater relative risk for subfertility compared to women with higher selenium concentrations. Estimated probabilities of subfertility based on multivariate adjusted models corresponded to an absolute difference of 6.3% when comparing low to high selenium concentrations (Figure 2). There were no associations between copper and zinc concentrations and risk for subfertility. Excluding women requiring ART, Appendix A also shows that women with lower selenium concentrations were at a 1.60 (1.06–2.41) greater relative risk for subfertility compared to women with higher selenium concentrations. When using study population tertiles rather than laboratory reference ranges, there were no associations between any of the trace elements and subfertility (Appendix A).

## 4. Discussion

This study provides novel insights into associations between trace element concentrations and TTP. We demonstrate that lower maternal plasma zinc and selenium concentrations were associated with an approximate one month longer TTP. Lower selenium concentrations were also associated with a 46% greater risk of subfertility, an absolute risk difference of 7% between subfertile and fertile women. Given the high prevalence of identified selenium and zinc deficiencies among this cohort, these finding could have important implications for improving fertility and warrant confirmation in subsequent studies, particularly where recruitment occurs prior to pregnancy. 

Selenium is an essential trace mineral, mainly found in seafood, poultry, eggs, and organ meats [28,29]. The most commonly used measures of selenium status are plasma and serum selenium concentrations, which reflect recent dietary selenium intake [30]. Functional measures of selenium status include the glutathione peroxidases and selenoprotein P which play important roles in antioxidant defence, formation of thyroid hormones, and DNA synthesis, all of which impact fertility and reproduction [8,10]. While there have been studies demonstrating that low selenium status during pregnancy is associated with pregnancy complications [31], roles of selenium in periconceptual events such as oocyte development, fertilisation, and implantation, have not been addressed to a large extent. However, there are reports of its potential importance in follicle growth and maturation [10]. The importance of selenium in male fertility has been extensively studied in humans and animals, particularly in regard to the biosynthesis of testosterone but also potentially in sperm motility [10]. In our study, the demonstration that low maternal plasma selenium concentrations associate with a longer time to pregnancy and a 46% greater risk of subfertility, after adjusting for a range of maternal and paternal factors, lends additional support for its role in female reproduction. Furthermore, in our supplementary analysis, even after excluding women requiring ART, lower selenium concentrations were still associated with a longer TTP and a 60% greater risk of infertility, in the final adjusted model. Additionally, we observed that selenium was not consumed in many multivitamin supplements across the fertile and subfertile groups. In restricting the analysis to only those who conceived naturally, we addressed any potential medical intervention bias, which occurs if the exposure of interest is also associated with the likelihood that couples would seek medical care for infertility. The mechanisms relating selenium to periconception health, particularly in women, require further investigation. Substantiating its effect on fertility may inform recommendations to increase maternal dietary selenium intake in women planning a pregnancy.

Women who had circulating zinc concentrations in the lowest reference range in early pregnancy took 0.6 months longer to conceive, compared with women in the middle tertile, after adjusting for maternal and paternal factors. Results were similar when excluding women requiring ART. Whilst the difference in TTP we observed only appears small, it is similar to the median difference in TTP with decreasing levels of fruit consumption and increasing levels of fast food consumption that we recently reported in the larger, international SCOPE cohort [19]. Diet is the main factor that determines zinc status [32]. In the United States and Australia, an additional 2–4 mg/day of zinc is recommended during pregnancy compared to non-pregnant women [28,33]. Fortunately many women across developed countries meet their country specific recommendations [34], suggesting adequate stores during pregnancy. Importantly, we show that at least half of all women consumed zinc in their multivitamin supplement. Zinc plays key roles in gene transcription, protein synthesis, and many other cellular processes including both antioxidant and pro-oxidant actions. Based on limited studies in women, the impact of maternal zinc concentrations on reproductive health prior to conception is inconclusive [35]. Studies in men have shown established roles for zinc in the synthesis of male sex hormones, sperm production and motility [10]. Furthermore, animal studies implicate zinc deficiency in impaired implantation [36], but also abnormal ovarian development, ovarian follicular growth, and oocyte maturation [10,36]. It is evident that zinc is required for key reproductive processes in animals and in men, and it is important in relation to adverse pregnancy outcomes in women. Whilst deficiencies in dietary zinc intake are not generally a problem in women in high income countries, a further understanding of whether, and how circulating zinc in women aids conception at the level of the oocyte and early embryo will help support our results on conception.

We found no association between maternal copper concentrations and TTP across any of the adjusted models. Copper is widely distributed in foods including organ meats, seafood, nuts and seeds, and it is biologically important for oxidative processes, energy metabolism, defence against free radicals, and for iron transport. Animal studies have shown copper deficiency in mice reduces fertilisation rates and oocyte recovery rate, but also heart and brain defects in the rat [36]. Using the same Adelaide SCOPE cohort, it was recently reported that women with lower plasma copper concentrations were protected against risk for any pregnancy complication when compared with women with high plasma copper [22], raising the possibility that copper may be more important for placentation rather than conception. In our cohort, only 26% and 44% of fertile and subfertile women, respectively, consumed copper in their multivitamin, but it is unclear whether this low percentage remarkably influenced the results. Further studies are required to determine the necessity of copper for early embryonic development and fertility in humans, and how oxidative damage from copper deficiency or toxicity may contribute to the occurrence of developmental defects in offspring.

Strengths of this study include the large sample size, detailed collection of maternal and paternal factors and information on the method of conception. The population was a community cohort of low risk nulliparous women, most of whom did not require any fertility treatment. In terms of study design, Jukic et al. highlight the strength of retrospective TTP studies in their ability to achieve a sample that is representative of the target population (i.e., women planning pregnancy), whereas prospective TTP studies that require the recruitment of highly motivated couples, introducing the potential for response and planning bias [37]. Limitations include the assessment of trace elements which were measured at 14–16 weeks’ gestation, rather than pre-pregnancy, thus we do not know whether such measurements truly reflect those at the time women were trying to conceive. Notably, the use of non-fasting samples for assessment of zinc, selenium and copper have recently been shown to be similar to fasting samples [38]. Potential limitations of retrospective TTP studies have been well described by us [19] and others [17,18] and include the potential for planning bias, medical intervention bias, truncation bias and behaviour change bias. We did not have data on menstrual cycle length which is often utilised alongside defining TTP. However, TTP is considered easier to remember in retrospective studies than cycle length [39], and previous studies have demonstrated that correcting for menstrual patterns had no impact on effect estimates [40]. Finally, long term recall of TTP can only be roughly estimated compared to prospective studies, however, the use of self-reported time to pregnancy is consistent with that of previous studies, supporting the use of retrospective questionnaires to be accurate in the assessment of TTP in fertile and subfertile couples [18], and show good agreement with prospective recall of TTP [37].

## 5. Conclusions

Lower maternal plasma zinc and selenium concentrations were associated with longer TTP, and lower selenium concentrations were also associated with a greater risk for subfertility. Our work points towards pre-conception care, specifically with a focus on increasing trace micronutrient intake, which may support fertility. Further research supporting our work is required, particularly regarding pre-conception assessment of circulating trace elements but also pre-conception micronutrient intake both from diet and from supplemental intake. Further human studies are required to examine relationships between circulating trace elements and fertility, but also to determine how circulating trace elements correlate with the follicular fluid microenvironment. Mechanistic studies are necessary to determine how circulating trace elements influence oocyte and embryo quality and development.

## Figures and Tables

**Figure 1 nutrients-11-01609-f001:**
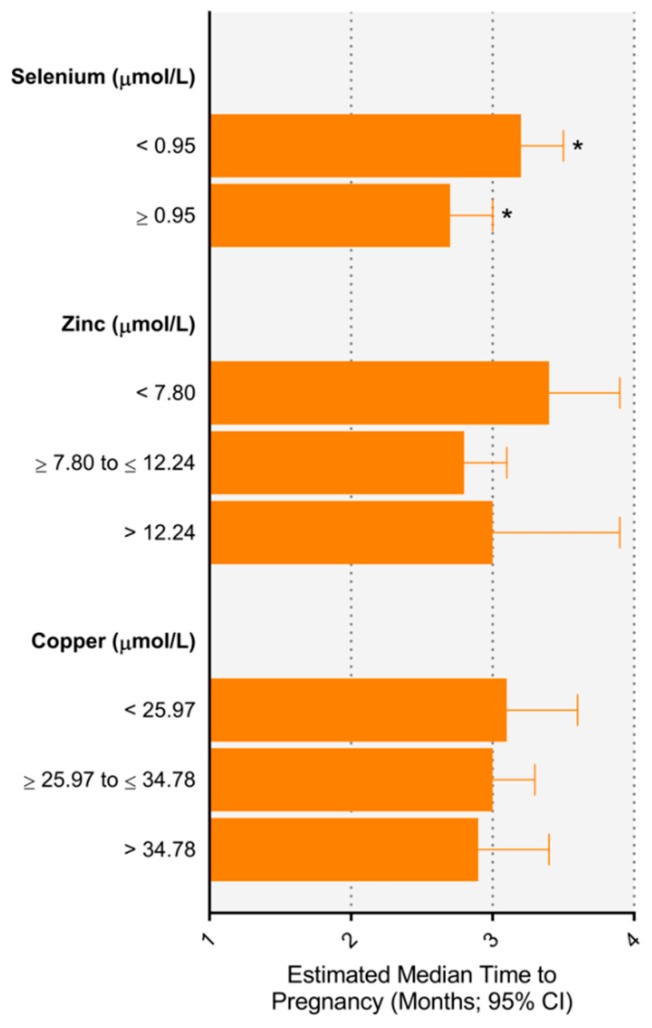
Estimated median time to pregnancy according to maternal trace element concentration. * *p* < 0.05 between groups.

**Figure 2 nutrients-11-01609-f002:**
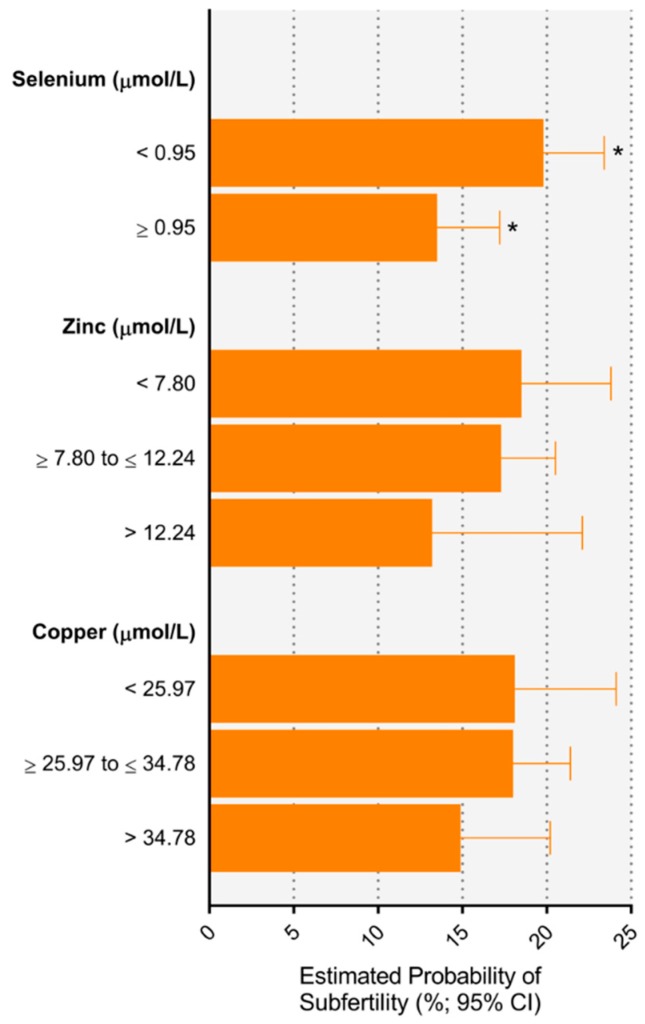
Estimated probability of subfertility according to maternal trace element concentration. * *p* < 0.05 between groups.

**Table 1 nutrients-11-01609-t001:** Characteristics of the study population (*n* = 1060).

Maternal Characteristics	Fertile	Subfertile
	**<12 months**	**≥12 months**
	894 (84.3%)	166 (15.7%)
Age (years), mean (SD)	23.4 (5.0)	25.4 (5.3)
Maternal age, ≥35 years, n (%)	20 (2.2%)	9 (5.4%)
Body mass index (kg/m2), mean (SD)	26.7 (6.3)	28.6 (7.3)
Socioeconomic index, mean (SD)	27.5 (10.2)	29.3 (11.0)
Ethnicity, n (%)		
Caucasian	822 (91.6)	149 (89.8)
Other	72 (8.1)	17 (10.2)
Trace element concentration		
Copper (µmol/L), mean (SD)	30.3 (5.4)	30.5 (6.1)
Zinc (µmol/L), mean (SD)	9.4 (2.2)	9.16 (2.7)
Selenium (µmol/L), mean (SD)	73 (12)	71 (11)
Frequency of sexual intercourse prior to pregnancy ^a^	18.1 (16.7)	15.4 (16.1)
Pre-pregnancy alcohol intake, yes (n %)	485 (54.3)	69 (41.6)
Pre-pregnancy smoking, yes (n %)	363 (40.6)	63 (38.0)
Pre-pregnancy food group intake (n %)		
Fast food, never	58 (8.3)	10 (6.9)
Fruit, ≥3/day	63 (7.1)	6 (3.6)
Green leafy vegetables, ≥1/day	227 (25.4)	45 (27.1)
Fish, ≥1/week	327 (36.6)	74 (44.6)
Multivitamin use in first trimester, yes (n %)	531 (59.5)	117 (70.5)
Multivitamin containing copper (n %)	234 (26.2%)	73 (44.0%)
Multivitamin containing selenium (n %)	16 (1.8%)	3 (1.8%)
Multivitamin containing zinc (n %)	458 (51.2%)	104 (62.7%)
Paternal characteristics (*n* = 930)		
Age (years), mean (SD)	26.7 (6.5)	28.7 (6.1)
Body mass index (kg/m^2^), mean (SD)	26.9 (5.1)	28.1 (5.5)
Missing, *n* = 99		

^a^ Frequency of sexual intercourse, per month, in the three months prior to conception, with the biological father of the baby. SD, standard deviation

**Table 2 nutrients-11-01609-t002:** Association between plasma trace element concentration measured at 15 ± 1 weeks’ gestation and time to pregnancy.

Trace Element	Concentration (μmol/L) ^a^	N	%	Unadjusted Time Ratio	Adjusted Time Ratio (95% CI)		
**Selenium**	<0.95	634	59.8%	1.10 (0.96–1.26)	1.14 (0.99–1.30)	1.17 (1.00–1.37)	1.19 (1.01–1.40)
	≥0.95	426	40.2%	1	1	1	1
**Zinc**	<7.80	237	22.4%	1.24 (1.05–1.46)	1.19 (1.01–1.41)	1.17 (0.98–1.40)	1.20 (0.99–1.44)
	≥7.80 to ≤12.24	734	69.3%	1	1	1	1
	>12.24	88	8.3%	1.14 (0.89–1.46)	1.17 (0.92–1.49)	1.08 (0.80–1.47)	1.05 (0.77–1.44)
**Copper**	<25.97	202		1.03 (0.87–1.22)	1.05 (0.89–1.24)	1.04 (0.86–1.25)	1.04 (0.86–1.26)
	≥25.97 to ≤34.78	506		1	1	1	1
	>34.78	186		1.03 (0.87–1.23)	0.98 (0.82–1.17)	0.94 (0.76–1.16)	0.97 (0.78–1.21)

^a^ Reference ranges from Abbassi-Ghanavati M. et al. [24]. ^b^ Adjusted for maternal age, maternal body mass index, ethnicity, socioeconomic status, plasma C-reactive protein, pre-pregnancy alcohol consumption, pre-pregnancy smoking status, frequency of sexual intercourse prior to conception, multivitamin use in first trimester, and trace element of interest in multivitamin. ^c^ Adjusted for b plus intake of fast food, green leafy vegetables, fruit, and fish in the one month prior to conception. ^d^ Adjusted for c plus paternal age and paternal body mass index.

**Table 3 nutrients-11-01609-t003:** Risk for subfertility (>12 months to conceive) according to plasma trace element concentration measured at 15 ± 1 weeks’ gestation.

Trace Element	Concentration (μmol/L) ^a^	N	n (%)	Unadjusted RR	Model 1 ^b^	Model 2 ^c^	Model 3 ^d^
**Selenium**	<0.95	634	113 (17.8)	1.43 (1.06–1.94)	1.52 (1.13–2.05)	1.44 (1.04–1.98)	1.46 (1.06–2.03)
	≥0.95	426	53 (12.4)	1	1	1	1
**Zinc**	<7.80	237	44 (18.6)	1.25 (0.91–1.72)	1.13 (0.82–1.56)	1.04 (0.74–1.45)	1.07 (0.76–1.50)
	≥7.80 to ≤12.24	734	109 (14.9)	1	1	1	1
	>12.24	88	13 (14.8)	0.99 (0.58–1.69)	1.07 (0.63–1.81)	0.88 (0.46–1.68)	0.0.76 (0.38–1.53)
**Copper**	<25.97	202	38 (15.8)	1.02 (0.72–1.44)	1.10 (0.77–1.58)	0.97 (0.67–1.40)	1.01 (0.69–1.47)
	≥25.97 to ≤34.78	506	93 (15.5)	1	1	1	1
	>34.78	186	35 (15.8)	1.02 (0.71–1.46)	0.88 (0.61–1.25)	0.79 (0.53–1.18)	0.83 (0.55–1.25)

Results displayed as relative risk (RR) and 95% CI. ^a^ Reference ranges from Abbassi-Ghanavati M. et al. [24]. ^b^ Adjusted for maternal age, maternal BMI, ethnicity, socioeconomic status, plasma C-reactive protein, pre-pregnancy alcohol consumption, pre-pregnancy smoking status, frequency of sexual intercourse prior to conception, multivitamin use in first trimester, and trace element of interest in multivitamin. ^c^ Adjusted for b plus intake of fast food, green leafy vegetables, fruit, and fish, in the one month prior to conception. ^d^ Adjusted for c plus paternal age and paternal body mass index.

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
