# Peer review of "Maternal Selenium, Copper and Zinc Concentrations in Early Pregnancy, and the Association with Fertility"

_nutrients, 2019, doi:10.3390/nu11071609_

Reviewer 1 Report

in my opinion the manuscript is well written and the authors answered completely to my questions

Reviewer 2 Report

This paper presents interesting data, it is well designed and described. It is resubmitted manuscript and Authors corrected it according to the Reviewers' comments in the former submission. I have only one question:

- "Materials and methods" "Trace element samples" page 3: Authors wrote that they used in plasma digestion 0% HNO3 ; is it mistake? could they explain ?

This manuscript is a resubmission of an earlier submission. The following is a list of the peer review reports and author responses from that submission.

Round  1

Reviewer 1 Report

This is a well-written and interesting paper.

 A few points:

 Should ‘subfertility’ be further defined by age category?

 Biochemical analysis of zinc, selenium and copper: more details would be helpful on methods. Zinc samples can be contaminated by dust and are influenced by time of last meal, processing methods, and fasting/non-fasting state. See paper by Arsenault et al: https://www.ncbi.nlm.nih.gov/pubmed/20978526 The details on these factors should be reported in full in methods/analysis section.

 Were values for zinc, copper and selenium measured in duplicate?

 “published pregnancy and laboratory values for second trimester of pregnancy” on line 99 – please provide a reference for these values.

 Zinc is a negative acute phase reactant and serum/plasma levels are influenced by inflammation – can this be accounted for? (should be for selenium and zinc as well). See paper by Diana et al: https://www.ncbi.nlm.nih.gov/pubmed/29189196

 ICP-MS reports in ng/mL – what conversation factors were used to calculate umol/L equivalent values – please report these.

 Multivitamin use in first trimester – differs from 60% vs 71% in the fertile and subfertile groups. Is there any further details on how this indicator is assessed (compliance?) is this factor adjusted for appropriately given the substantial differences. Lack of further data on this variable is a major limitation given the effect that micronutrient supplementation has on zinc, selenium and copper concentrations.

Author Response

REVIEWER 1

This is a well-written and interesting paper.

 A few points:

 Comment 1: Should ‘subfertility’ be further defined by age category?

 Response 1: This was a young cohort with an approximate age of 23-25 years of age. There was only a small percentage of women who were advanced age, 35 years and over. We have now included the number/percentage of women 35 years and over into Table 1.

 Comment 2: Biochemical analysis of zinc, selenium and copper: more details would be helpful on methods. Zinc samples can be contaminated by dust and are influenced by time of last meal, processing methods, and fasting/non-fasting state. See paper by Arsenault et al: https://www.ncbi.nlm.nih.gov/pubmed/20978526 The details on these factors should be reported in full in methods/analysis section.

 Response 2: Thankyou for this comment. We understand that laboratory samples can be contaminated by a range of external factors. Our samples were analysed by an accredited laboratory in South Australia, ensuring a contamination free environment. Because the blood samples were taken from pregnant women, it is not feasible to obtain a fasting sample

We have amended the paragraph to the following (page 3):

Non-fasting plasma was obtained from heparinised, venous blood samples taken at 15±1 weeks’ gestation. In order to measure trace element concentrations, plasma samples were digested in concentrated nitric acid (~70% HNO3) under pressure and at temperature as per our previous publication [15]. Prior to analysis, 250 μL of plasma was digested in concentrated nitric acid (0% HNO3) in sealed Teflon containers for approximately 48 h and then diluted. Samples were run alongside two internal standards: iridium and rhodium (Choice Analytical) at a concentration of 200 ppb and an 8-point calibration, including blank, was carried out between 0.01 μg/L and 100 μg/L. Copper, selenium and zinc concentrations were determined using inductively-coupled plasma mass spectrometry (ICP-MS) (Agilent 7700 ICP-MS; carried out by accredited CSIRO Analytical Services, South Australia). Laboratory surfaces were thoroughly cleaned to avoid contamination and all procedures were performed under conditions that ensured no trace mineral contamination. Samples were run in triplicate through the mass spectrometer. Concentration categories for each trace element were made in accordance with published pregnancy and laboratory studies values for second trimester of pregnancy, with reference points being the middle concentration range: copper (<25.97 25.97="" to="" 34.78="">34.78 umol/L); zinc (<7.80 7.80="" to="" 12.24="">12.24 umol/L); selenium (<0.95 0.95="" to="" 1.84="">1.84 umol/L) [16]. However, for selenium, no woman had a value over 1.84 µmol/L, thus the categories were amended to:<0.95 <umol/L vs. ≥0.95 unmol/L [reference value]).

Comment 3: Were values for zinc, copper and selenium measured in duplicate?

Response 3: We have included in the above paragraph that samples were run in triplicate.

Comment 4: “published pregnancy and laboratory values for second trimester of pregnancy” on line 99 – please provide a reference for these values.

Response 4: The reference was included at the end of the sentence [reference 16, Abbassi-Ghanavati, M.; Greer, L. G.; Cunningham, F. G. Pregnancy and laboratory studies: a reference table for clinicians. Obstet Gynecol. 2009, 114, 1326-1331]

Comment 5: Zinc is a negative acute phase reactant and serum/plasma levels are influenced by inflammation – can this be accounted for? (should be for selenium and zinc as well). See paper by Diana et al: https://www.ncbi.nlm.nih.gov/pubmed/29189196 

Response 5: Thankyou for this comment. We acknowledge that zinc is an acute phase reactant; and we are aware of studies showing CRP, as a marker of systemic inflammation, is associated with adverse pregnancy outcomes. We have now adjusted all of our analyses for plasma CRP measured at 15 weeks’ gestation. This did not significantly influence the results.

Comment 6: ICP-MS reports in ng/mL – what conversation factors were used to calculate umol/L equivalent values – please report these.

Response 6: We reported in SI units (umol/L) as per the journal guidelines. The conventional units (mg/L) from the mass spec were converted using St Vincents Syd Path conversion table factors: http://www.sydpath.stvincents.com.au/other/Conversions/ConversionMasterF3.htm

Zinc conventional units mg/dL x 0.153 to SI mmol/L;

Selenium conventional units mg/L x 0.0127 to SI mmol/L;

Copper conventional units mg/dL x 0.157 to SI mmol/L

Because we are reporting in SI units, we have included the following in the methods (trace element sample section) to convert to convention units: “To convert from SI units (mmol/L) to conventional units, the following equations can be used: zinc (mmol/L x 6.53 mg/dL); selenium (mmol/L x 78.74 mg/L); copper (mmol/L x 6.37 mg/dL)”.

Comment 7: Multivitamin use in first trimester – differs from 60% vs 71% in the fertile and subfertile groups. Is there any further details on how this indicator is assessed (compliance?) is this factor adjusted for appropriately given the substantial differences. Lack of further data on this variable is a major limitation given the effect that micronutrient supplementation has on zinc, selenium and copper concentrations.

Response 7: This was a prospective cohort study in pregnancy, and multivitamin supplementation compliance is not assessed. We acknowledge that multivitamin intake would increase blood concentrations of these nutrients, thereby decreasing the risk of infertility. However, our results do not suggest this, and this shows bias towards the null because ART/subfertility is not influencing plasma concentrations. We had adjusted for multivitamin intake (yes/no), but we now additionally adjust for whether the multivitamin included the particular trace element (i.e. in analyses assessing zinc concentrations, we adjust for whether zinc (yes/no) was contained in the multivitamin). In Table 1, we now include the n/% of women who consumed a multivitamin containing either copper, selenium, or zinc.

Reviewer 2 Report

I think that the manuscript is very interesting and well designed with a large sample size and is it innovative about the etiology of infertility. My mayor concerns regard:

1)    It would be interesting to divide the population in a subgroup of pregnant women who received assisted reproductive technologies and to assess in those women the study of infertility and ovaric function such as anti-mullerian hormone (AMH) and the relationship with levels of copper, zinc and selenium in order to avoid bias.

2)    Authors did not consider male factors of infertility I suggest to assess it, if it is possible

3)    Authors didn’t assess in the study the menstrual cycle before pregnancy as a factor linked to infertility. Pathology like polycystic ovary syndrome (PCOS) should be considered in one of the Statistical Models

4)    In the discussion or conclusions authors should explain and suggest recommendations to increase blood concentrations of zinc and selenium during pre-conception period in order to avoid infertility

5)     I think authors have to assess in the number of servings of food: red meat and legumes as beef is the most important source of zinc 

Author Response

REVIEWER 2

I think that the manuscript is very interesting and well designed with a large sample size and is it innovative about the etiology of infertility. My mayor concerns regard:

Comment 1: It would be interesting to divide the population in a subgroup of pregnant women who received assisted reproductive technologies and to assess in those women the study of infertility and ovaric function such as anti-mullerian hormone (AMH) and the relationship with levels of copper, zinc and selenium in order to avoid bias.

Response 1: Thankyou for this comment. We now include an additional two supplementary tables (supplemental tables 1 and 3) excluding women requiring ART, assessing trace element concentrations and time to pregnancy and risk for subfertility. We cannot assess subfertility in women requiring ART because all women undergoing ART are already considered subfertile. We have commented on the supplementary tables in the results.

Comment 2: Authors did not consider male factors of infertility I suggest to assess it, if it is possible.

Response 3: In our third adjusted model, we present covariates using paternal age and paternal BMI. These are very important factors in relation to infertility.

Comment 3: Authors didn’t assess in the study the menstrual cycle before pregnancy as a factor linked to infertility. Pathology like polycystic ovary syndrome (PCOS) should be considered in one of the Statistical Models

Response 3: Thankyou for this comment. We are aware that including menstrual cycle length is a useful exposure in terms of time to pregnancy. We did not collect this data and we have included a sentence on this in the last paragraph of the discussion (page 10):

“We also did not have data on menstrual cycle length which is often utilized alongside defining TTP. However, TTP is considered easier to remember in retrospective studies than cycle length (Olsen, Bolumar et al. 1997), and previous studies have demonstrated that correcting for menstrual patterns had no impact on effect estimates (Hassan and Killick 2004).”

Comment 4: In the discussion or conclusions authors should explain and suggest recommendations to increase blood concentrations of zinc and selenium during pre-conception period in order to avoid infertility.

Response 4: We have included a sentence in the conclusion to state: “Our work points towards pre-conception care, specifically with a focus on increasing trace micronutrient intake, which may support time to conception.”

Comment 5: I think authors have to assess in the number of servings of food: red meat and legumes as beef is the most important source of zinc

Response 5: Thankyou for this comment. Unfortunately we did not assess intake of these foods but we do acknowledge they are good sources of zinc, and future research may benefit from assessing these foods on time to pregnancy.

Reviewer 3 Report

 The manuscript presents interesting data. Generally, the study is rather well designed and described. However, there are some issues in this manuscript:

 1.       Title is too general: in my opinion instead of “trace element” should be “plasma zinc and selenium”.

2.       Material and methods: in exclusion criteria Authors wrote: “we excluded women (…) where blood samples were unavailable for trace element analysis” what is mean? Why these samples were not available for analysis? In exclusion criteria there is nothing about mineral supplements or fortified food intake ; daily minerals intake with supplements/mineral water/fortified products  may affect  mineral status in pregnant women.

3.       Assessment of outcome: estimation of TTP is not very precise; there are a lot of factors which may influence on TTP (e.g. frequently of sexual intercourse and time of sex  in relation to menstrual cycle).

4.       Why mineral intake and mineral supplements intake were not used  in models ? it is very important factor; e.g.  only multivitamin use in first trimester was included in models.

5.       Results: in description and in tables and  figures there is lack information about significance.

6.       Discussion is too general. Obtained results are rather poor explained. Discussion should also include two very important observations: 1. “women who took longer to conceive tended to have a lower frequency of sexual intercourse” 2. “ a higher percentage of (women who took longer to conceive) consumed a multivitamin in the first trimester”, these factors may influence on obtained results ; e.g.  interaction between vitamins and minerals may affect mineral status.

Author Response

REVIEWER 3

The manuscript presents interesting data. Generally, the study is rather well designed and described. However, there are some issues in this manuscript:

Comment 1: Title is too general: in my opinion instead of “trace element” should be “plasma zinc and selenium”.

Response 1: We have amended the title to: Maternal selenium, copper and zinc concentrations in early pregnancy, and the association with fertility.

Comment 2: Material and methods: in exclusion criteria Authors wrote: “we excluded women (…) where blood samples were unavailable for trace element analysis” what is mean? Why these samples were not available for analysis? In exclusion criteria there is nothing about mineral supplements or fortified food intake ; daily minerals intake with supplements/mineral water/fortified products  may affect  mineral status in pregnant women.

Response 2: There were some samples missing in our biobank of blood samples, and therefore we did not include these women because we had no data on their exposure (i.e. trace element concentrations).

Comment 3: Assessment of outcome: estimation of TTP is not very precise; there are a lot of factors which may influence on TTP (e.g. frequently of sexual intercourse and time of sex  in relation to menstrual cycle).

Response 3: Thank you for this comment. We believe that TTP is a valuable outcome measure for assessment of fecundability and we had included in the discussion the strengths of using time to pregnancy as an outcome variables: “In terms of study design, Jukic et al. highlights the strength of retrospective TTP studies in their ability to achieve a sample that is representative of the target population (i.e. women planning pregnancy), as opposed to prospective TTP studies that require the recruitment of highly motivated couples, introducing the potential for response and planning bias (4).

We have now included a limitation for this outcome in the discussion (page 9):

Long term recall of time to pregnancy can only be roughly estimated compared to prospective studies, however, the use of self-reported time to pregnancy is consistent with that of previous studies, supporting the use of retrospective questionnaires to be accurate in assessment of TTP in fertile and sub-fertile couples [36] and show good agreement with prospective recall of TTP [34].”

Reviewer 4 Report

The data obtained by the authors have enormous potential but they have not been properly used. A few erroneous assumptions have been made:1.      the concentration of Se, Zn and Cu during the 15th week of pregnancy is a reflection of the status of these elements in the period preceding fertilization - this assumption can not be accepted because physiologically in pregnancy there is a lower concentration of e.g. selenium. In addition, the determination of the concentration of these elements in the plasma reflects the short-term status of saturation. It depends on the current way of feeding, the occurrence of vomiting typical of the first trimester of pregnancy and the use of vitamin and mineral supplements dedicated to pregnant women.2.      the fertility of the couple depends only on the condition of the future mother – if we want to talk about fertility, it should be analyzed the health condition - in this case the status of Se, Zn and Cu - in both women and men.3.      fruits, green leafy vegetables, fast food and fish are a good nutritional source of Se, Zn and Cu - among these sources, only fish are rich in these elements. It is puzzling why the consumption of meat, organ meats, cereals, eggs and legumes was not taken into account. 
               Apart from the incorrect assumptions, the authors in the Introduction section omitted the role of the discussed elements in male fertility. It is also not clear why 3 models for analysis were specified.               In Results section, line 165, the authors state "Compared to reference value, women who had lower selenium concentration (<0.95 umol="" l="" vs.="" 0.95="" had="" -="" what="" value="" is="" the="" reference="" for="" in="" line="" authors="" state="" higher="" levels=""> 0.97 umol / L vs. ≤0.89 to ≥0.97 umol / L) were associated with .." - such ranges of values do not appear in any of the tables or figures. In table 2 the numerical values for selenium are given for the units ug/L and not umol/L.               In the Discussion section, the authors did not refer to the created models  at all. 

Author Response

(Rest of reviewer 3 comments)

Comment 4: Why mineral intake and mineral supplements intake were not used  in models ? it is very important factor; e.g.  only multivitamin use in first trimester was included in models.

 Response 4: As per the previous comment on multivitamin supplementation (Reviewer 1, comment 7), we now adjust for multivitamin intake and adjust each analysis for the particular trace element.

Comment 5: Results: in description and in tables and figures there is lack information about significance.

 Response 5: In the text and tables, statistical significance is reported as the 95% confidence intervals. In the figures, we now include P values where differences between groups exist.

Comment 6: Discussion is too general. Obtained results are rather poor explained. Discussion should also include two very important observations: 1. “women who took longer to conceive tended to have a lower frequency of sexual intercourse” 2. “ a higher percentage of (women who took longer to conceive) consumed a multivitamin in the first trimester”, these factors may influence on obtained results ; e.g.  interaction between vitamins and minerals may affect mineral status.

Response 6: Thankyou for this comment. Whilst we do not feel that the discussion is “too general”, it is suitable for readers of the Nutrients journal given the nutritional relationships to micronutrient concentrations. Additional, because there is a lack of human data on the same topic, it is difficult to describe mechanisms. We have however, elaborated on our results in both the selenium and zinc sections. Specifically, the section on selenium we now include:

“Furthermore, in our supplementary analysis, even after excluding women requiring ART, lower selenium concentrations were still associated with a longer TTP and a 60% greater risk for infertility. Additionally, we observed that selenium was not consumed in many multivitamin supplements across the fertile and subfertile groups. Restricting the analysis to only those who conceived naturally, we address any potential medical intervention bias, and that women receiving fertility treatments was not influencing plasma concentrations.”

In Table 1, we do report that women who took longer to conceive tended to have a lower frequency of sexual intercourse, however, this equates to only 1 less time per month of sexual intercourse. Therefore, it is unlikely to influence the results severely and we do not know if this was consistent behaviour contributing to subfertility.

REVIEWER 4

The data obtained by the authors have enormous potential but they have not been properly used. A few erroneous assumptions have been made:

Comment 1. The concentration of Se, Zn and Cu during the 15th week of pregnancy is a reflection of the status of these elements in the period preceding fertilization - this assumption can not be accepted because physiologically in pregnancy there is a lower concentration of e.g. selenium. In addition, the determination of the concentration of these elements in the plasma reflects the short-term status of saturation. It depends on the current way of feeding, the occurrence of vomiting typical of the first trimester of pregnancy and the use of vitamin and mineral supplements dedicated to pregnant women.

Response 1: Thankyou for this comment. We used reference ranges at the time of gestation which takes into account physiological changes in pregnancy. We have acknowledged this as a limitation in the discussion and recognise that 15 weeks’ gestation is a proxy for pre-pregnancy.

Comment 2: The fertility of the couple depends only on the condition of the future mother – if we want to talk about fertility, it should be analyzed the health condition - in this case the status of Se, Zn and Cu - in both women and men.

Response 2: We agree that any measure of fertility applies to couples, not individuals, and that certain exposures of the man influence their fertility. However, the data were from a prospective cohort from 10 years ago and we do not have these samples to assess reproductive function or semen quality.

Comment 3: Fruits, green leafy vegetables, fast food and fish are a good nutritional source of Se, Zn and Cu - among these sources, only fish are rich in these elements. It is puzzling why the consumption of meat, organ meats, cereals, eggs and legumes was not taken into account.

Response 3: We only had limited dietary intake data and did not assess the consumption of meat and legumes, and we have recently published showing a higher intake of fruit and a lower intake of fast food associated with time to pregnancy (Grieger JA et al, Human Reproduction, 2018). We believe this would be an interesting area of research for future studies on this topic.

Comment 4: Apart from the incorrect assumptions, the authors in the Introduction section omitted the role of the discussed elements in male fertility.

Response 4: We have now included a sentence on male fertility in the introduction “and there are established roles for selenium, zinc, and copper in relation to male fertility [10, 11]” and we had previously made comment to male fertility in the discussion.

Comment 5: It is also not clear why 3 models for analysis were specified.

Response 5: We indicated in the methods that 3 separate models were included. Three models were included so we could adjust for additional covariates in models 2 and 3, to see whether the effect remained when including maternal diet (model 2) and paternal age and BMI (model 3). We know paternal age and BMI associate with fertility, and we have recently published showing maternal diet associates with fertility (Grieger JA et al, Human Reproduction, 2018), so we included these in different models beyond that of usual standard covariates.

Comment 6: In Results section, line 165, the authors state "Compared to reference value, women who had lower selenium concentration (<0.95 umol="" l="" vs.="" 0.95="" had="" -="" what="" value="" is="" the="" reference="" for="" in="" line="" authors="" state="" higher="" levels="">0.97 umol / L vs. ≤0.89 to ≥0.97 umol / L) were associated with .." - such ranges of values do not appear in any of the tables or figures.

Response 6: We apologise for not included these in the methods. We now state:

Concentration categories for each trace element were made in accordance with published pregnancy and laboratory studies values for second trimester of pregnancy, with reference values being the middle concentration values: copper (<25.97 mmol/L, ≥25.97 to ≤34.78 mmol/L, >34.78 mmol/L); zinc (<7.80 mmol/L, ≥7.80 to ≤12.24 mmol/L, >12.24 mmol/L); selenium (<0.95 mmol/L, ≥0.95 to 1.84 mmol/L, >1.84 mmol/L) [18]. However for selenium, no woman who had a value over 1.84 µmol/L, thus the categories were amended to:<0.95 mmol/L vs. ≥0.95 [reference value]).

Comment 7: In table 2 the numerical values for selenium are given for the units ug/L and not umol/L.

 Response 7: Thankyou for picking this up. There was a slight error in the calculation that we have now fixed, and units are umol/L.

Comment 8: In the Discussion section, the authors did not refer to the created models at all.

Response 8: We had made a comment on the adjusted models in the selenium paragraph but we have now made mention of these in the zinc and copper paragraphs.

Round  2

Reviewer 3 Report

I accept the improved version of the manuscript.

Reviewer 4 Report

Unfortunately, the most important assumptions of the research still speak against it.

Comment 1. The concentration of Se, Zn and Cu during the 15th week of pregnancy is a reflection of the status of these elements in the period preceding fertilization - this assumption can not be accepted because physiologically in pregnancy there is a lower  concentration  of  e.g.  selenium. In  addition,  the  determination  of  the concentration  of  these  elements  in  the  plasma  reflects  the  short-term  status  of saturation. It depends on the current way of feeding, the occurrence of vomiting typical  of  the  first  trimester  of  pregnancy  and  the  use  of  vitamin  and  mineral supplements dedicated to pregnant women.

Response 1: Thankyou for this comment. We used reference ranges at the time of gestation which takes into account physiological changes in pregnancy. We have acknowledged this as a limitation in the discussion and recognise that 15 weeks’ gestation is a proxy for pre-pregnancy.

The assumption that the degree of saturation with elements measured at the beginning of pregnancy corresponds to the pre-pregnancy state is untrue, even if it is referred to reference values for pregnancy. This is a non-chronological setting of the effect and cause. The correlations obtained may be completely accidental, it is as if we correlated fish consumption among Eskimos with the risk of cancer among Australians - here also statistically correlations can be demonstrated. If the authors measured the elements content in the hair or nails  (which illustrates the long-term state of saturation), we could approximately assume that the obtained results correspond to the saturation status with elements from the pre-pregnancy period.

Comment 2: The fertility of the couple depends only on the condition of the future mother – if we want to talk about fertility, it should be analyzed the health condition - in this case the status of Se, Zn and Cu - in both women and men.

Response 2: We agree that any measure of fertility applies to couples, not individuals, and that certain exposures of the man influence their fertility. However, the data were from a prospective cohort from 10 years ago and we do not have these samples to assess reproductive function or semen quality.

In this case, it was necessary to select only the women on whose side, within the examined couple, infertility was clearly identified.

Comment 3: Fruits, green leafy vegetables, fast food and fish are a good nutritional source of Se, Zn and Cu - among these sources, only fish are rich in these elements. It is puzzling why the consumption of meat, organ meats, cereals, eggs and legumes was not taken into account.   Response  3:  We  only  had  limited  dietary  intake  data  and  did  not  assess  the consumption  of  meat  and  legumes,  and  we  have  recently  published  showing  a higher  intake  of  fruit  and  a  lower  intake  of  fast  food  associated  with  time  to pregnancy (Grieger JA et al, Human Reproduction, 2018). We believe this would be an interesting area of research for future studies on this topic.

I am not surprised at all that the low intake of highly-processed foods and the high intake of fruits as a source of vitamin C, beta-carotene and fiber positively influenced fertility in the studied group of women. However, the aim of this study is the effect of the plasma concentration of selected elements on fertility and therefore the intake of good nutritional sources of these elements should be investigated, or in thecase of  absence of data, this factor should not be considered. And thus, the models analyzed should also be changed.

Based on the above arguments, I believe that the research should be carried out again based on correct assumptions.